# Sentinel Lymph Node Biopsy in Surgical Staging for High-Risk Groups of Endometrial Carcinoma Patients

**DOI:** 10.3390/ijerph19063716

**Published:** 2022-03-21

**Authors:** Antonio Raffone, Diego Raimondo, Antonio Travaglino, Giulia Rovero, Manuela Maletta, Ivano Raimondo, Marco Petrillo, Giampiero Capobianco, Paolo Casadio, Renato Seracchioli, Antonio Mollo

**Affiliations:** 1Gynecology and Obstetrics Unit, Department of Neuroscience, Reproductive Sciences and Dentistry, School of Medicine, University of Naples Federico II, 80131 Naples, Italy; anton.raffone@gmail.com; 2Division of Gynaecology and Human Reproduction Physiopathology, Department of Medical and Surgical Sciences (DIMEC), IRCCS Azienda Ospedaliero-Universitaria di Bologna S. Orsola Hospital, University of Bologna, Via Massarenti 13, 40138 Bologna, Italy; giulia.rovero91@gmail.com (G.R.); manuela.maletta@studio.unibo.it (M.M.); paolo.casadio@aosp.it (P.C.); renato.seracchioli@unbo.it (R.S.); 3Anatomic Pathology Unit, Department of Advanced Biomedical Sciences, School of Medicine, University of Naples Federico II, 80131 Naples, Italy; 4Gynecologic and Obstetric Unit, Department of Medical, Surgical and Experimental Sciences, University of Sassari, 07100 Sassari, Italy; pwraimo@gmail.com (I.R.); marco.petrillo@gmail.com (M.P.); capobia@uniss.it (G.C.); 5Gynecology and Obstetrics Unit, Department of Medicine, Surgery and Dentistry “Schola Medica Salernitana”, University of Salerno, 84081 Baronissi, Italy; amollo@unisa.it

**Keywords:** endometrium, risk assessment, lymphadenectomy, lymph node dissection, mapping, treatment

## Abstract

Background: In endometrial carcinoma (EC) patients, sentinel lymph node (SLN) biopsy has shown the potential to reduce post-operative morbidity and long-term complications, and to improve the detection of low-volume metastasis through ultrastaging. However, while it has shown high sensitivity and feasibility in low-risk EC patient groups, its role in high-risk groups is still unclear. Aim: To assess the role of SLN biopsy through the cervical injection of indocyanine green (ICG) in high-risk groups of early-stage EC patients. Materials and methods: Seven electronic databases were searched from their inception to February 2021 for studies that allowed data extraction about detection rate and accuracy of SLN biopsy through the cervical injection of ICG in high-risk groups of early-stage EC patients. We calculated pooled sensitivity, false negative (FN) rate, detection rate of SLN per hemipelvis (DRh), detection rate of SLN per patients (DRp), and bilateral detection rate of SLN (DRb), with 95% confidence interval (CI). Results: Five observational cohort studies (three prospective and two retrospective) assessing 578 high risk EC patients were included. SLN biopsy sensitivity in detecting EC metastasis was 0.90 (95% CI: 0.03–0.95). FN rate was 2.8% (95% CI: 0.6–11.6%). DRh was 88.4% (95% CI: 86–90.5%), DRp was 96.6% (95% CI: 94.7–97.8%), and DRb was 80% (95% CI: 75.4–83.9). Conclusion: SLN biopsy through ICG cervical injection may be routinely adopted instead of systematic pelvic and para-aortic lymphadenectomy in surgical staging for high-risk groups of early-stage EC patients, as well as in low-risk groups.

## 1. Introduction

Endometrial carcinoma (EC) is the most common gynecological malignancy in developed countries, showing an increase even higher in number of deaths than in incidence in the last decades [1,2,3,4,5,6,7,8,9]. The increase in number of deaths appears due to an inaccurate management of patients for adjuvant treatment [10]. To date, adjuvant treatment is based on pathological and molecular features of specimens from surgical staging [11]. In fact, surgical staging for apparent uterine-confined ECs consists of total hysterectomy with bilateral salpingo–oophorectomy and nodal assessment, including pelvic lymphadenectomy with or without para-aortic lymphadenectomy [11,12]. Although lymph node staging has shown utility in directing adjuvant treatment [13,14], it carries the risk of increased morbidity, including lymphoedema, lymphocyst formation, and nerve injury. Moreover, it is technically difficult to perform in the obese population, which represents a large proportion of patients with ECs [15].

In order to overcome these limitations, in 2014, the National Comprehensive Cancer Network (NCCN) guidelines approved the sentinel lymph node (SLN) biopsy as an alternative to systematic lymphadenectomy for the staging of apparent uterine-confined ECs in selected cases [12]. SLN biopsy has shown the potential to significantly reduce the risk of post-operative morbidity and long-term complications [16]. Additionally, it may be associated with a more intensive pathologic assessment (i.e., ultrastaging), with the advantage of detecting low-volume metastasis, which could be missed by standard histological examination [17,18].

Despite these advantages, some issues needing to be addressed for SLN biopsy in ECs include tracers to be used, preferred injection sites, and applicability in the different risk groups of EC patients.

Among the several tracers and injection sites proposed for SLN mapping [19], the cervical injection of fluorescent dye indocyanine green (ICG) has shown the highest bilateral pelvic detection rate and is recommended at the present time [20,21,22].

Regarding applicability in patient risk groups, while SLN biopsy has shown high sensitivity and feasibility in low-risk EC patients [10,19,23], its role in high-risk EC patients is still unclear.

The aim of this study was to assess the role of SLN biopsy through the cervical injection of ICG in high-risk groups of early-stage EC patients.

## 2. Materials and Methods

### 2.1. Study Protocol

An *a priori* study protocol was built for all study stages, including search strategy, study selection, risk of bias evaluation, data extraction, and analysis. Two authors independently concluded all stages, discussing disagreements with all authors.

The Preferred Reporting Item for Systematic Reviews and Meta-analyses (PRISMA) and the Synthesizing Evidence from Diagnostic Accuracy Tests (SEDATE) guidelines and checklist [24,25] were followed for reporting our study.

### 2.2. Search and Selection of Studies

MEDLINE, Google Scholar, Web of Sciences, Scopus, Cochrane Library, EMBASE, and ClinicalTrial.gov were searched as electronic databases from their inception to February 2021. The following text words were alternatively combined in several searches: “endometr*”, “cancer”, “carcinoma”, “neoplasia”, “malignancy”, “tumor”, “uter*”; “corpus uteri”; “sentinel lymph node biopsy”; “sentinel lymph node dissection”; “SLND”; “lymphatic mapping”; “SLN”; “ICG”; “indocyanine green”; “dye”; “lymph node dissection”; “LND”; “lymphadenectomy”; “staging”; “ultrastaging”; “algorithm”; “lymph”. Searches also included references screening from full-text assessed studies.

We included all peer-reviewed studies that allowed data extraction about detection rate and accuracy of SLN biopsy through the cervical injection of ICG in high-risk groups who were early-stage EC patients. In particular, we excluded studies considering:Patients who did not undergo bilateral pelvic systematic lymphadenectomy with/without para-aortic lymphadenectomy as a reference;Exclusively low-risk groups of EC patients;Injection site of ICG different from uterine cervix;Dye different from ICG.

Reviews and case reports were also excluded.

### 2.3. Risk of Bias within Studies Evaluation

The revised Quality Assessment of Diagnostic Accuracy Studies (QUADAS-2) was followed for evaluating the risk of bias within studies [26]. In detail, all included studies were judged in 4 domains related to the risk of bias: (1) patient selection (i.e., if all eligible patients were included); (2) index test (i.e., if SLN biopsy and histological examination was performed as recommended [11,12]); (3) reference standard (i.e., if systematic lymphadenectomy was performed as recommended [11,12]); (4) flow and timing (i.e., if all patients included in the analysis were assessed with the same index test and reference standard). Each domain was judged at “low risk,” “high risk” or “unclear risk” of bias based on if data were “reported and adequate”, “reported but inadequate”, or “not reported”, respectively.

### 2.4. Data Extraction

Data were extracted without modification of original data according to the PICO (Population, Intervention, Comparator, Outcomes) items [24]. In particular, the “Population” for our study was high-risk EC patients. The “Intervention” was EC metastasis in specimens from the SLN biopsy. The “Comparator” was EC metastasis in specimens from systematic lymphadenectomy. The “Outcomes” were sensitivity of SLN biopsy in detecting EC metastasis, false negatives (FN) rate, detection rate of SLN per hemipelvis (DRh, i.e., the proportion of hemipelves with SLN detected), detection rate of SLN per patient (DRp, i.e., the proportion of patients with at least one SLN detected), and bilateral detection rate of SLN (DRb, i.e., the proportion of patients with SLN detected in both hemipelves).

For each included study, two by two contingency tables were built based on two qualitative variables:EC metastasis in specimens from SLN biopsy (index test), dichotomized as “absent” and “present”;EC metastasis in specimens from systematic lymphadenectomy (reference standard), dichotomized as “absent” and “present”.

### 2.5. Data Analysis

We calculated sensitivity, FN rate, DRh, DRp, and DRb with 95% confidence interval (CI) as the individual and pooled estimate, and graphically reported values on forest plots.

Sensitivity analysis and FN rate were based on patients with bilateral SLN detection. In particular, patients with at least one hemipelvis not mapped were excluded from analysis.

Patients with EC metastasis in specimens from the SLN biopsy were considered as true positive (the presence of metastasis in specimens from systematic lymphadenectomy is unnecessary to define true positives because SLN may be the only location for EC metastasis).

Patients without EC metastasis in specimens from both SLN biopsy and systematic lymphadenectomy were considered as true negatives.

Patients without EC metastasis in specimens from SLN biopsy but showing EC metastasis in specimens from systematic lymphadenectomy were considered as false negatives.

False positives were not evaluable because if SLN is positive, lymph node metastasis is certain (i.e., SLN may be the only location for EC metastasis).

Statistical heterogeneity among studies was assessed through the Higgins’ inconsistency index (I^2^). It was judged as null for I^2^ = 0%, minimal for 0% < I^2^ ≤ 25%, low for 25 < I^2^ ≤ 50%, moderate for 50 < I^2^ ≤ 75%, and high for I^2^ > 75%, as previously reported [27,28].

The random effect model of DerSimonian and Laird was adopted for all analyses independently from the statistical heterogeneity, as recommended for meta-analysis of diagnostic accuracy by the SEDATE guidelines [25].

Meta-DiSc version 1.4 (Clinical Biostatistics Unit, Ramon y Cajal Hospital, Madrid, Spain) and Review Manager version 5.4 (The Nordic Cochrane Centre, The Cochrane Collaboration, 2014, Copenhagen, Denmark) were used as software for analysis.

## 3. Results

### 3.1. Study Selection

Electronic searches led to 9120 articles. Duplicate removal, title screening, and abstract screening led to 6028, 193, 24 articles, respectively. These articles underwent full-text assessment, which led five articles to be included in the qualitative and quantitative analysis [29,30,31,32,33]. The study selection flow is graphically reported in Appendix A.

### 3.2. Study and Patients’ Characteristics

All included studies were observational cohort studies: three were prospective [30,32,33] and two were retrospective [29,31]. A total of 684 EC patients were assessed, 578 (84.5%) of whom were with high-risk ECs. The definition of high-risk groups in the included studies is reported in Appendix A.

Patient mean age and body mass index ranged from 53–71 years and 24.8–27.5 kg/m^2^, respectively (Appendix A).

EC histotype was endometrioid in 57.6% of cases, serous in 24.6%, carcinosarcoma in 6.6%, clear cell in 6.1%, and undifferentiated in 1.6% (Appendix A). ECs had International Federation of Gynecology and Obstetrics (FIGO) stage IA in 40.9% of cases, IB in 33.8%, II in 4.4%, IIIA in 1.6%, IIIB in 0.4%, IIIC1 in 11.4%, IIIC2 in 6.6%, and IV in 0.9% (Appendix A).

Surgical staging was laparoscopic in two studies [29,32], robotic in one study [30], laparoscopic or robotic in one study [33], and unspecified in the other study [31]. Pathological ultrastaging was performed in 4 studies [29,30,32,33]. Details about ICG injection technique are reported in Appendix A.

Details about sentinel lymph node biopsy and systematic lymphadenectomy are reported for each included study in Appendix A.

### 3.3. Risk of Bias within Studies

All included studies were judged at low risk of bias in the “patient selection”, “reference standard”, and “flow and timing” domains.

In the “index test” domain, two studies were judged at unclear risk of bias: one because histological examination of specimen from SLN biopsy did not include ultrastaging [31], and the other because it did not evaluate SLN in para-aortic region [30].

Risk of bias within studies evaluation was graphically summarized, as shown in Appendix A.

### 3.4. Meta-Analysis

SLN biopsy sensitivity in detecting EC metastasis was 0.90 (95% CI: 0.03–0.95; I^2^: 76.6%; Figure 1). FN rate was 2.8% (95% CI: 0.6–11.6%; I^2^: 79.8%; Figure 2).

DRh was 88.4% (95% CI: 86–90.5%; I^2^: 20.1%; Figure 3), DRp was 96.6% (95% CI: 94.7–97.8%; I^2^: 0%%; Figure 4), and DRb was 80% (95% CI: 75.4–83.9; I^2^: 28.9%; Figure 5). Detection rates were calculated without considering ICG reinjection since only one study performed it [30].

## 4. Discussion

### 4.1. Main Findings and Interpretation

This study shows that SLN biopsy through the cervical injection of ICG has a high sensitivity in detecting EC metastasis in high-risk groups of early-stage EC patients. Moreover, such technique has a low FN rate and high DRh, DRp and DRb.

SLN biopsy emerged in order to reduce peri-operative morbidity and long-term lymphatic complications of systematic lymphadenectomy, and to overcome technical difficulties in performing the procedure. The first successful case of SLN biopsy was historically described in 1977, with a lymphangiography of the penis. [34] Since then, SLN biopsy techniques have been studied and developed for several solid malignancies, among them, breast cancer and melanoma [35,36]. In gynecologic oncology, SLN biopsy first reached agreement for patients with vulvar cancer. Subsequently, it was also encouraging in the management of cervical and ECs [10].

In fact, since 2014, the National Comprehensive Cancer Network (NCNN) approved SLN biopsy as an alternative to systematic lymphadenectomy for the staging of apparent uterine-confined ECs in selected cases [12]. However, while feasibility and sensitivity in detecting EC metastasis was proven to be enough in low-risk EC patient groups [37], its applicability in high-risk groups is still unclear to date. This might be due to the higher rate of LN metastasis for high-risk groups of EC patients, which imposes, according to previous studies, caution in avoiding systematic lymphadenectomy in these patients. [38]

Indeed, although the last updated ESGO/ESTRO/ESP guidelines state that SLN biopsy is an acceptable alternative to systematic lymphadenectomy for lymph node staging in high-intermediate/high-risk EC patients, the level of evidence for the recommendation is III, highlighting the need for further evaluation [11]. Such evaluation should include DR, FN rate, and sensitivity in detecting EC metastasis. In fact, in order to replace systematic lymphadenectomy, SLN biopsy must have high DR and sensitivity and a low FN rate. In our study, we found that SLN biopsy had high sensitivity in high-risk EC patients, similarly to that reported for EC patients at both low risk (96%) [19] and unspecified risk (93%) [39]. Moreover, we found even higher DRh, DRp, and DRb when compared to those in low-risk ECs (88.4% vs. 66%, 96.6% vs. 81%, 80% and 50%, respectively). This might reflect the inclusion in the analysis of studies adopting not-optimized SLN biopsy techniques. In fact, studies adopting dyes different from ICG and injection sites different from the cervix were pooled together [19]. Nevertheless, ICG cervical injection is the preferred injection technique at present because of its high bilateral pelvic success rate and reproducibility [40], its para-aortic detection rate is still debated. In fact, Laios et al. highlighted an association between dye injected site and location of detected SLN [41]. In detail, they showed a case of para-aortic SLN detection subsequent to exclusive ICG injection in the uterine fundus rather than in the uterine cervix [41]. Thus, dye injection sites remain to be further investigated especially for the application of SLN biopsy in the surgical staging for high-risk EC patients.

Our data would indicate the applicability of SLN biopsy also in high-risk EC patients. Regardless, the need for a strict adherence to the Memorial Sloan Kettering SLN algorithm is remarkable [42]. This algorithm includes retroperitoneal evaluation with excision of any suspicious enlarged nodes regardless of mapping and side-specific pelvic and para-aortic lymphadenectomy in case of an unmapped hemipelvis [11,12]. Although additional studies are necessary, it appears conceivable that SLN biopsy might direct adjuvant management in high-risk EC women similarly to low-risk EC patients.

However, despite the applicability, the usefulness of SLN biopsy in EC patients should be re-discussed after the innovative findings about molecular classification from The Cancer Genome ATLAS Research Network (TCGA) [3,43]. In particular, TCGA has shown that ECs can be classified in four prognostic groups based on molecular signature, with the potential for reducing the current under- and over-treatment of EC patients [3,43]. These groups seem to show different lymph node involvement rates and prognosis, with different need for adjuvant treatment, independently from the classic pathological factors [3,27,44,45,46,47,48,49]. In detail, DNA polymerase epsilon (POLE)-mutated and p53-wild type patients have shown a 0% and 4% rate of lymph node involvement, respectively, suggesting the chance to preoperatively choose for avoiding nodal assessment [3,47]. On the other hand, p53- abnormal patients show the worst prognosis, requiring adjuvant therapy independently from nodal status [3]. Future studies are necessary to investigate this issue.

### 4.2. Strengths and Limitations

To the best of our knowledge, this study may be the first systematic review and meta-analysis to investigate the role of SLN biopsy through the cervical injection of ICG in high-risk groups of early-stage EC patients. Our findings are supported by an overall low risk of bias within the studies, as shown in the risk of bias within studies evaluation.

However, our study may be affected by several limitations. In particular, a major limitation may be the fact that not all included patients underwent systematic para-aortic lymphadenectomy, but patients were assigned to systematic para-aortic staging based on technical feasibility. This may affect our results even more than those about low-risk groups of EC patients due to the higher incidence of isolated para-aortic EC metastasis in high-risk groups (1–3% vs. 5%) [50]. Therefore, we could have underestimated FN rate and overestimated sensitivity of SLN biopsy.

Another limitation may be the impossibility to assess SLN biopsy sensitivity and FN rate per hemipelvis due to the absence of specific extractable data from four of the five included studies [29,30,31,32]. However, we restricted such analyses to patients with bilateral mapping, which was the best surrogate for hemipelvis assessment.

Finally, a limitation of our study may be the low number of inclusive studies (n = 5) and the absence of RCTs. In fact, the inclusion of exclusively observational studies may affect the level of evidence for recommendations from our systematic review and meta-analysis.

## 5. Conclusions

SLN biopsy through ICG cervical injection may be routinely adopted instead of systematic pelvic and para-aortic lymphadenectomy in the surgical staging for high-risk groups of early-stage EC patients, as well as in low-risk groups.

Further studies, with particular regard to RCTs, are encouraged in the field.

## Figures and Tables

**Figure 1 ijerph-19-03716-f001:**
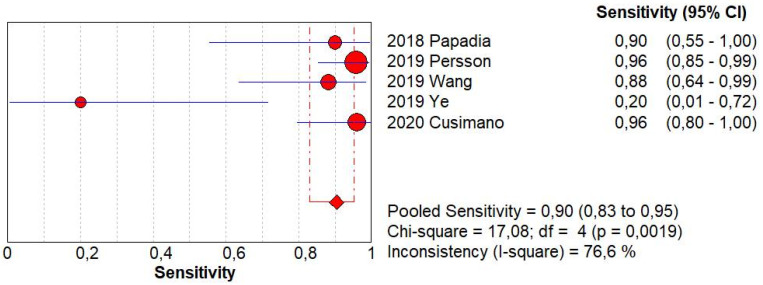
Forest plots of individual studies and pooled sensitivity of sentinel lymph node biopsy in detecting endometrial carcinoma metastasis.

**Figure 2 ijerph-19-03716-f002:**
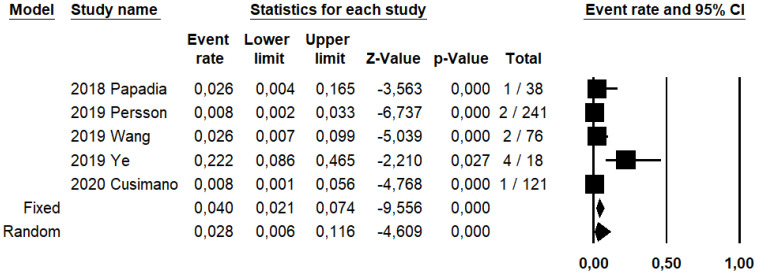
Forest plots of individual studies and pooled false negative rate of sentinel lymph node biopsy in detecting endometrial carcinoma metastasis.

**Figure 3 ijerph-19-03716-f003:**
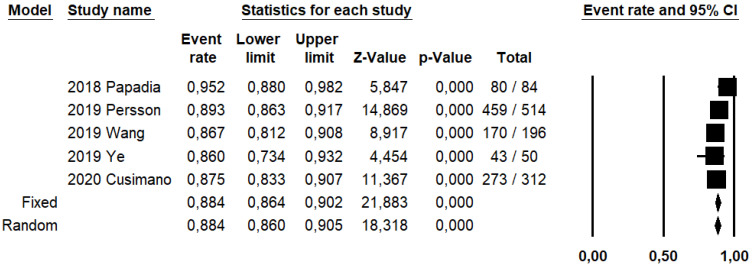
Forest plots of individual studies and pooled detection rate of sentinel lymph node per hemipelvis.

**Figure 4 ijerph-19-03716-f004:**
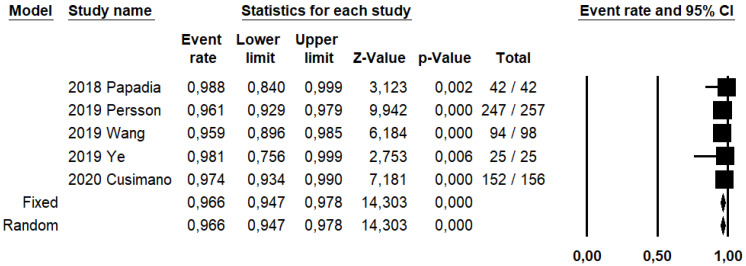
Forest plots of individual studies and pooled detection rate of sentinel lymph node per patient.

**Figure 5 ijerph-19-03716-f005:**
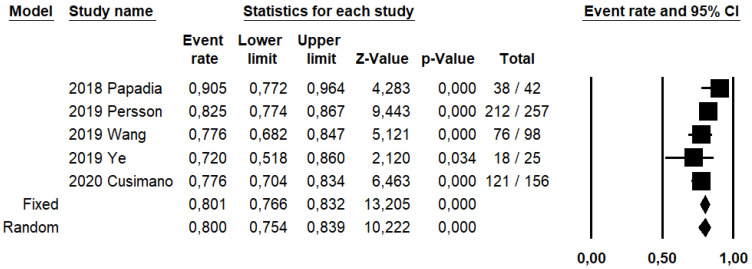
Forest plots of individual studies and pooled bilateral detection rate of sentinel lymph node.

## Data Availability

Not applicable.

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
