# Peer review of "Sentinel Lymph Node Biopsy in Surgical Staging for High-Risk Groups of Endometrial Carcinoma Patients"

_ijerph, 2022, doi:10.3390/ijerph19063716_

Round 1

Reviewer 1 Report

This is a well conducted systematic review on sentinel LN in early stage HGS endometrial ca. It does add to the literature and I recommend publication 

I think this systematic review is well designed and conducted by following the standard PRISMA methodology. The Pooled estimates are done correctly and the metaanalysis is well performed. The studies and bias effects are thoroughly considered.    The main weakness which I recommend authors to consider is to illustrate the difficulty generalizing the findings as there were only 5 studies and non were a randomised controlled trial. Therefore the level of evidence will not be level I, I think still level II with C level of recommendation RCT is required before we draw a line on safety and efficacy

Author Response

  1. a) Author response: We thank the Reviewer for the kind comments and suggestions. We discussed the low number of inclusive studies (n=5) and the absence of RCTs as a limitation of our study which may affect the level of evidence. Moreover, we highlighted the need for future studies and RCTs in the Conclusion section of the revised manuscript.
  2. b) Location: Page 9, lines 340-343, 348

Reviewer 2 Report

The study was motivated by the recent NCNN and ESTRO guidelines for uterine cancer suggesting further evaluation of the role of SLN biopsy in high risk early stage cancer. Undoubtedly, the quality of the presentation and the scientific soundness is high. Nevertheless, the authors need to furhter state why the role of SLN biopsy in high risk uterine cancers has been unclear so far. There is a reason for caution, which is that 50% of high risk endometrial cancers have metastases, hence SLN biopsies should not routinely performed in those groups according to Ballester M et al. (SENTI-ENDO STUDY, Lancet Oncology, 2011). The authors also do not seem to have quoted the meta-analysis of 26 studies by Kang S et al, (Gynecol Oncol, 2011), which reported similar sensitivity between low and high risk groups, albeit the numbers were small, in addition to significant heterogeneity. This should be critically appraised in the discussion section. Rather than adopting a de facto applicability of SLN biopsy (with ICG injections and ultrastaging as correctly discussed, irrespective of the slow implementation of the molecular analysis for endometrial cancer), the study should continue to encourage high risk EC patient recruitment in clinical trials investigating the role of SLN surgery, especially those with non-endometrioid tumors. In addition of the accuracy of SLN biopsies clearly stated in this study, the authors should mention the work needed to establish patterns of lymph node involvement in these tumors and whether SLN biopsy can be used to direct adjuvant treatment by allowing the omission of adjuvant treatment in those women who are node negative. 

Finally, the adherence to the principles of SLN mapping for EC staging in the absence of demonstrable metastasis by imaging or at exploration should be stated in the discussion section. That is, SLN mapping with ultrastaging by ICG cervical injection and performance of site specific nodal dissection if mapping fails or grossly enlarged/suspicious lymph nodes irrespective of mapping. The comment about the current pragtice regarding para-aortic lymph nodes can be potentially expanded. It remains debatable how good ICG injection can be for para-aortic SLN detection as studies employing multiple fluorescent dues, enabled visualisation of independently draining lymphatic paths in women with endometrial cancer (Laios A et al, BMC Research notes, 2015).

Author Response

Comment #1

The study was motivated by the recent NCNN and ESTRO guidelines for uterine cancer suggesting further evaluation of the role of SLN biopsy in high risk early stage cancer. Undoubtedly, the quality of the presentation and the scientific soundness is high. Nevertheless, the authors need to furhter state why the role of SLN biopsy in high risk uterine cancers has been unclear so far. There is a reason for caution, which is that 50% of high risk endometrial cancers have metastases, hence SLN biopsies should not routinely performed in those groups according to Ballester M et al. (SENTI-ENDO STUDY, Lancet Oncology, 2011).

  1. a) Author response: We thank the Reviewer for the suggestion. We clarified this aspect in the Discussion section of the revised manuscript.
  2. b) Location: Pages 7,8; lines 236-238.

Comment #2

The authors also do not seem to have quoted the meta-analysis of 26 studies by Kang S et al, (Gynecol Oncol, 2011), which reported similar sensitivity between low and high risk groups, albeit the numbers were small, in addition to significant heterogeneity. This should be critically appraised in the discussion section.

  1. a) Author response: We thank the Reviewer for the interesting comment. We discussed this finding in the revised manuscript, as suggested.
  2. b) Location: Page 8; line 246.

Comment #3

Rather than adopting a de facto applicability of SLN biopsy (with ICG injections and ultrastaging as correctly discussed, irrespective of the slow implementation of the molecular analysis for endometrial cancer), the study should continue to encourage high risk EC patient recruitment in clinical trials investigating the role of SLN surgery, especially those with non-endometrioid tumors.

  1. a) Author response: We thank the Reviewer for the advice. We discussed the absence of clinical trials as a limitation of our study which may affect the level of evidence. Moreover, we encourage highlighted clinical trials in the field in the Conclusion section of the revised manuscript.
  2. b) Location: Page 9, line 348.

Comment #4

In addition of the accuracy of SLN biopsies clearly stated in this study, the authors should mention the work needed to establish patterns of lymph node involvement in these tumors and whether SLN biopsy can be used to direct adjuvant treatment by allowing the omission of adjuvant treatment in those women who are node negative.

  1. a) Author response: We thank the Reviewer for the interesting comment. Although additional studies are necessary, we hypothesized that SLN biopsy might direct adjuvant management in high risk EC women similarly to low risk EC patients.
  2. b) Location: Page 8; lines 263-265.

Comment #5

Finally, the adherence to the principles of SLN mapping for EC staging in the absence of demonstrable metastasis by imaging or at exploration should be stated in the discussion section. That is, SLN mapping with ultrastaging by ICG cervical injection and performance of site specific nodal dissection if mapping fails or grossly enlarged/suspicious lymph nodes irrespective of mapping.

  1. a) Author response: We thank the Reviewer for the comment and suggestion. We specified the importance of adherence to the Memorial Sloan Kettering SLN algorithm when SLN biopsy is used for surgical staging in EC patient in the Discussion section of the revised manuscript.
  2. b) Location: Page 8, lines 260-263.

Comment #6

The comment about the current practice regarding para-aortic lymph nodes can be potentially expanded. It remains debatable how good ICG injection can be for para-aortic SLN detection as studies employing multiple fluorescent dues, enabled visualisation of independently draining lymphatic paths in women with endometrial cancer (Laios A et al, BMC Research notes, 2015).

  1. a) Author response: We thank the Reviewer for the advice. We discussed the need for further investigation of dye injection site especially for high risk EC patients based on the suggested findings.
  2. b) Location: Page 8; lines 251-258.
